Recommended nitrogen rates and the verification of effects based on leaf SPAD readings of rice

Hou Wenfeng 1
Shen Juan 1
Xu Weitao 1
Khan Muhammad Rizwan 2
Wang Yin 1
Zhou Xue 1
Gao Qiang 1
Murtaza Behzad 3
Zhang Zhongqing 447000257@qq.com 1
1 Key Laboratory of Soil Resource Sustainable Utilization for Jilin Province Commodity Grain Bases/College of Resources and Environmental Science, Jilin Agricultural University, Changchun, Jilin , Changchun , China
2 Soil and Environmental Sciences Division, Nuclear Institute for Agriculture and Biology (NIAB) , Faisalabad , Pakistan
3 Department of Environmental Sciences, COMSATS University Islamabad, Vehari Campus , Vehari , Pakistan
Kalaji Hazem
Electronic publication date: 2021 Aug 26
Publication date: 2021
Volume: 9
Electronic Location ID: e12107
Received 2021 Jan 5; Accepted 2021 Aug 11
Copyright: ©2021 Hou et al.
Copyright year: 2021
Copyright holder: Hou et al.
License: This is an open access article distributed under the terms of the Creative Commons Attribution License, which permits unrestricted use, distribution, reproduction and adaptation in any medium and for any purpose provided that it is properly attributed. For attribution, the original author(s), title, publication source (PeerJ) and either DOI or URL of the article must be cited.
License URL: https://creativecommons.org/licenses/by/4.0/

Keywords: SPAD reading, Nitrogen diagnosis, Non-destructive, Chlorophyll, Leaf greenness

Funding: Scientific Research Start-up Fund of Jilin Agricultural University 202023330 National Key Research and Development Program of China 2016YFD0200101 Science and Technology Project of the 13th Five-Year Plan of Jilin Provincial Department of Education JJKH20190908KJ This work was supported by the Scientific Research Start-up Fund of Jilin Agricultural University (202023330); the National Key Research and Development Program of China (2016YFD0200101); and the Science and Technology Project of the 13th Five-Year Plan of Jilin Provincial Department of Education (JJKH20190908KJ). The funders had no role in study design, data collection and analysis, decision to publish, or preparation of the manuscript.

==============================
Modern rice production systems need a reliable, easy-to-use, efficient, and environmentally-friendly method to determine plant nitrogen (N) status , predict grain yield, and optimize N management. We conducted field experiments to determine the influence of different N rates on Soil Plant Analysis Development (SPAD) readings of rice leaves. We also performed field validations to evaluate the grain yield and N use efficiency under recommended N rates. Our results showed that leaf SPAD readings increased as N rates increased. We applied the recommended N based on the relationships between the N rates and leaf SPAD readings at the tillering and booting stages. The recommended N decreased N rates and improved N use efficiency without sacrificing grain yield. When compared to farmer practices (FP), the recommended N rates of optimization (OPT) decreased by 5.8% and 10.0%, respectively. In comparison with FP, the N agronomic efficiency of OPT increased by 5.8 and 10.0% while the partial factor productivity of N increased by 6.0 and 14.2%, respectively. The SPAD meter may be a reliable tool to analyze the N in rice, estimate real-time N fertilization, and improve N use efficiency.

Introduction

Rice (Oryza sativa L.) is one of the most important cereal crops and food sources in the world. Rice yields must be increased in order to meet the food demands of ever-growing populations (Guo et al., 2019) and synthetic fertilizers, particularly nitrogen (N), play a vital role in improving rice yields. N is arguably the most important limiting factor, aside from water, for rice production. Global rice yield significantly increased with rising N rates. In current agricultural systems, most farmers apply N in excess relative to the actual crop’s needs to harvest more grain. This practice leads to low N use efficiency due to high N losses through runoff, denitrification, leaching, volatilization, and a high risk of environmental contamination (Zeng et al., 2012; Liu et al., 2013; Bodirsky et al., 2014; Xiong et al., 2015). The average N use in China’s rice production has been reported to be significantly higher than the global average (Chen, Huang & Tang, 2011). It is imperative to optimize N fertilizer use to prevent damage to crops and the environment, improve N use efficiency, and maintain production system sustainability. The use of controlled-release N fertilizers, nitrification inhibitors, and deep conventional urea placement are methods that have been tested to prevent N losses and improve N use efficiency. N fertilizer may be reduced by 12–50% without sacrificing grain yield (Qiao et al., 2012; Wang et al., 2012).

It is important to accurately measure leaf N content because it plays a crucial role in the growth and development of rice. Chemical analysis of leaf N content has been used to monitor N status and estimate the demand for N fertilizer (Ziadi et al., 2008; Tian et al., 2011). The current method is destructive and time-consuming so finding an indirect method to measure leaf N content is of great importance (Uddling et al., 2007; Yuan et al., 2016). Leaf N content is strongly correlated with chlorophyll content (Wang et al., 2014; Yang et al., 2014; Kalaji et al., 2017), thus leaves’ spectral characteristics may be used to guide N fertilizer use. Soil Plant Analysis Development (SPAD) readings can be used to assess the in situ leaf N status based on light transmitted through leaves. The use of SPAD readings to monitor rice N status has been used to improve grain yield and N use efficiency over the past two decades (Khurana et al., 2007; Huang et al., 2008; Li et al., 2009; Hou et al., 2020). Available N affects the chlorophyll content of the leaf, as N is one of the key elements of chlorophyll. Errecart et al. (2012) observed a close correlation between chlorophyll content and leaf N concentration. SPAD may reflect the plant leaves’ N status and is simple, quick, reliable, and harmless (Li et al., 2009; Xiong et al., 2015; Akhter et al., 2016; Yuan et al., 2016). There has been a significant correlation between SPAD readings for dry mass leaves’ N content and grain yield (Ramesh et al., 2002; Parvizi et al., 2004), indicating that SPAD readings may help determine optimum N application rates.

We sought to: (1) estimate the relationship between leaf SPAD readings and recommended N rates; (2) investigate the yield effects and N use efficiency of recommended N rates based on leaf SPAD readings.

Materials and Methods

Experimental site

We conducted two field experiments and two verification experiments (45.09°N, 124.92°E; 45.13°N, 124.89°E) in 2008 and 2009 involving different N rates in the Songyuan, Jilin Province, which is one of the major regions for japonica rice production in China. Both testing sites had black soils. The physicochemical properties of the experimental soils were pH 8.3 and pH 7.2 (soil: water = 1: 2.5), organic matter content 12.6 g kg−1 and 18.7 g kg−1, total N 1.4 g kg−1 and 2.1 g kg−1, Olsen-P 12.2 mg kg−1 and 9.0 mg kg−1, and readily available K 98.4 mg kg−1 and 49.0 mg kg−1, respectively.

Weather parameters

Songyuan has a mid-temperate continental monsoon climate. The annual average temperature, annual average sunshine, frost-free period, and annual precipitation in the rice-growing seasons were 5.1 °C, 2,878 h, 140 days, and 450 mm, respectively. Figure 1 shows the daily temperature and precipitation during the rice-growing season. The total precipitation during the rice-growing season was 387 and 218 mm in each field.

Figure 1 Temperature and precipitation during the rice-growing stage.

Experimental design and managements

In 2008, we conducted two field trials using seven nitrogen (N) rates (0, 45, 90, 135, 180, 225, and 270 kg N ha−1) in two different fields in the same location. We used the same planting pattern in both fields. We then used a randomized complete block design with three replicates. Nitrogen (N), phosphorus (P), and potassium (K) were applied as urea (46% N), superphosphate (12%, P2O5), and potassium chloride (60% K2O), respectively. Nitrogen was applied in three doses: 1/2 dose in the form of basal fertilizer 1 day before transplanting, 1/4 dose in the first topdressing 7 days after transplanting, and 1/4 dose as the second top dressing at the panicle initiation stage. Demonstrative experiments were conducted in 2009 close to where we experimented with different N rates. The two demonstration experiments were comprised of three treatments: zero N (CK), farmer practice (FP, 180 kg N ha−1), and optimization (OPT, N rate was recommended based on leaf SPAD readings).

Phosphorous was applied at a uniform rate of 90 kg P2O5 ha−1 in each experimental plot as a basal dressing one day before transplanting. We applied 90 kg K2O ha−1 in each experimental plot: 70% was applied as a basal dressing 1 day before transplanting; 30% was applied as a top dressing at the panicle initiation stage. The fertilizers were distributed to each plot and then mixed evenly by a rake. Each plot measured 30 m2 (6 m ×5 m).

We used the hybrid rice Jigeng 88 with a growth period of 145 days in these experiments. This rice variety is widely planted in the Northeast China Plain basin. Seedlings were raised in a seedbed. Seeds were sown on May 3rd and May 22nd, and seedlings were transplanted on June 3rd and June 29th in 2008 and 2009, respectively, at the four-leaf-stage. We planted a single plant per hill; hills were spaced 30 cm ×20 cm and followed the local agricultural technology department’s recommended practices. Rice seeds and plants were treated with the same fungicides, insecticides, and herbicides to avoid yield losses. We irrigated and performed other agronomic practices according to high-yield and standard protocols (Zhang et al., 2009).

Soil sample collection and analysis

Surface soil (0–20 cm) samples were randomly collected with a drag-type drill from five points in the experimental field two days before transplanting on a sunny day. The composite soil samples were air-dried, ground, and passed through one mm and 0.149 mm sieves to determine soil physicochemical characteristics. The soil pH, organic matter, total N, available P, and exchangeable K were determined following Bao (2000)’s method.

SPAD reading

A total of six plants placed in the center of each plot were marked after transplanting. A chlorophyll meter (Soil Plant Analysis Development, SPAD-502) was used for taking the SPAD reading measurements of the uppermost fully expanded leaves at the tillering stage, jointing stage, booting stage, heading stage, flowering stage, and filling stage. Three SPAD readings per leaf were taken around the midpoint and again 30 mm away on both sides of the midpoint. Each plot’s SPAD reading was determined by taking the average of 18 readings (Peng et al., 1993).

Plant harvest

Plants were harvested on September 15th and October 1st in two consecutive years. Each plot’s grain yield was determined from the 15 m2 area by measuring from the center of each plot and was adjusted to 14% moisture.

Calculations

Nitrogen use efficiency was expressed in terms of N agronomic efficiency (NAE) and partial factor productivity of N (PFPN), which were calculated as follows (Jian et al., 2014):

NAE (kg kg−1) = (grain yield of N-fertilized plot - grain yield of zero-N plot) / applied N rate.

PFPN (kg kg−1) = grain yield of N-fertilized plot / applied N rate.

Statistical analysis

All figures were drawn using Origin 8.0 software (Origin Lab, Massachusetts, USA). The means were separated using one-way analysis of variance with a Tukey test at a 5% probability level using SPSS 17.0 software (SPSS Inc., Chicago, USA).

Results

Effects of N rates on leaf SPAD readings

The leaf SPAD readings first increased and then decreased during the reproductive period (Fig. 2). Higher N rates significantly increased the leaf SPAD readings. The N0 treatment resulted in the lowest leaf SPAD readings, while N270 showed the highest leaf SPAD readings in both years. In 2008, N270 leaf SPAD readings increased by 31.5–42.8% in experiment 1 and by 27.2–56.2% in experiment 2 when compared with N0.

Figure 2 Temperature and precipitation during the rice-growing stage.

There was a significant positive linear relationship between N rates and leaf SPAD readings at the tillering and booting stages in both experiments (Fig. 3). The following linear correlations were found between N rates and leaf SPAD readings at the tillering stages in experiments 1 and 2, respectively:

Figure 3 Relationships between N rates with SPAD at tillering (A and B) and booting stage (C and D).

Y = 0.0833X + 30.4 (a)

Y = 0.1006X + 29.8 (b)

The following linear correlations were found between the N rates and leaf SPAD readings at the booting stages in experiments 1 and 2, respectively:

Y = 0.0568X + 30.5 (c)

Y = 0.0480X + 30.8 (d)

Grain yield

The grain yield quickly responded to N rates in both experiments (Fig. 4). An increase in N rates significantly improved the grain yield relative to the N0 treatment. However, further increases in N rates caused significantly different yield responses. The grain yield decreased when the N rate was higher than 235 kg ha−1 in experiment 1 (Fig. 4A). The relationship between the N rates and grain yield was unitary conic. In contrast, the grain yield increased with an increase in N rates and then became stagnant as the N rate rose beyond 195 kg ha−1 in experiment 2 (Fig. 4B).

Figure 4 (A–B) Effects of different N rates on grain yield of rice.

Recommended N rates at the tillering and booting stages

The recommended N rates at the tillering and booting stages based on leaf SPAD readings are shown in Tables 1 and 2, respectively. The N supply increased the leaf SPAD readings in experiments 1 and 2 by 21.3% and 22.7% at the tillering stage, and by 27.1% and 13.6% at booting stage, respectively, compared with CK. The optimum N rate was 235 kg ha−1 in experiment 1 and 195 kg ha−1 in experiment 2 in 2008 based on the relationship between N rates and grain yield (Fig. 4). Basal dosages in the OPT treatments were based on the N usage mode; in experiment 1 the dosage was 78 kg N ha−1, in experiment 2 the dosage was 65 kg N ha−1, and in the FP treatments the dosage was 90 kg N ha−1. At the tillering stage, the leaf SPAD readings in the OPT treatments were 33.0 in experiment 1 and 32.6 in experiment 2, respectively. The optimum leaf SPAD readings at the tillering stage in experiments 1 and 2 were 36.9 and 36.3, respectively. These findings were based on the relationship between the N rates and leaf SPAD readings at the tillering stage (Fig. 3A & b). N rate supply in experiments 1 and 2 were determined to be 12.0 kg ha−1 and 9.7 kg ha−1, respectively, for the unit increase in leaf SPAD readings. The recommended N rates at the tillering stage in experiment 1 and 2 were 46.8 and 35.9 kg ha−1, respectively (Table 1).

Table 1 Leaf SPAD readings and recommended N rates of the verification experiments at tillering stage.

Treatments	Experiment 1	Experiment 2	
	SPAD	N rate
kg N ha−1	SPAD	N rate
kg N ha−1	
CK	28.7 ± 0.6 c	0 ± 0 c	28.6 ± 0.9 c	0 ± 0 c	
FP	36.6 ± 1.4 a	45.0 ± 0.8 b	37.6 ± 1.7 a	45.0 ± 1.2 a	
OPT	33.0 ± 1.1 b	46.8 ± 0.9 a	32.6 ± 1.3 b	35.9 ± 1.0 b	

Table 2 Leaf SPAD readings and recommended N rates of the verification experiments at booting stage.

Treatments	Experiment 1	Experiment 2	
	SPAD	N rate
kg N ha−1	SPAD	N rate
kg N ha−1	
CK	28.6 ± 1.1 b	0 ± 0 b	30.9 ± 0.7 b	0 ± 0 c	
FP	35.9 ± 0.8 a	45.0 ± 0.4 a	35.5 ± 1.1 a	45.0 ± 1.0 b	
OPT	36.8 ± 1.2 a	44.7 ± 0.5 a	34.7 ± 0.9 a	51.6 ± 1.2 a	

Similarly, leaf SPAD readings of the OPT treatments at the booting stage were 36.8 in experiment 1 and 34.7 in experiment 2. The optimum SPAD readings in experiments 1 and 2 were 39.4 and 37.1, respectively, based on the relationship between the N rates and leaf SPAD readings at the booting stage (Figs. 3C & 3D). The N supply rates in experiments 1 and 2 were determined to be 17.2 and 21.5 kg ha−1, respectively, for the unit increase in leaf SPAD reading. Hence, the recommended N rates at the tillering stage in experiments 1 and 2 were 44.7 and 51.6 kg ha−1, respectively (Table 2).

Grain yield and N use efficiency

The grain yield and N use efficiency of the demonstration experiments are shown in Table 3. Compared with CK, the N supply increased grain yield by 347.1 and 102.5% in experiments 1 and 2, respectively. However, there was no significant difference between the yield of FP and OPT recorded. The recommended N rates on the basis of leaf SPAD readings were 169.5 and 152.5 kg ha−1 in experiments 1 and 2, respectively. Compared with FP, the recommended N rates of OPT decreased by 5.8% in experiment 1 and 15.3% in experiment 2. Moreover, the OPT’s NAE and PFPN were higher than FP’s in both experiments. The OPT’s NAE increased by 5.8 and 10.0% while PFPN increased by 6.0 and 14.2% compared to FP in experiments 1 and 2, respectively.

Table 3 Grain yield, N rates and N use efficiency of the verification experiments.

Treatments	Grain yield
kg ha−1	N rate
kg ha−1	NAE
kg kg−1	PFPN
kg kg−1	
Experiment 1	CK	1951 ± 136 b	0 ± 0 c	—	—	
FP	8733 ± 245 a	180.0 ± 0 a	37.7 ± 1.1 b	48.5 ± 0.9 b	
OPT	8713 ± 289 a	169.5 ± 1.6 b	39.9 ± 0.7 a	51.4 ± 0.7 a	
Experiment 2	CK	3394 ± 208 b	0 ± 0 c	—	—	
FP	6989 ± 368 a	180.0 ± 0 a	20.0 ± 0.8 b	38.8 ± 1.3 b	
OPT	6756 ± 404 a	152.5 ± 1.8 b	22.0 ± 1.1 a	44.3 ± 1.8 a	

Discussion

We determined N based on non-destructive leaf SPAD readings, which may help to decrease N rates. The recommended N rates based on leaf SPAD readings improved the N use efficiency of rice without sacrificing grain yield. SPAD readings gradually increased as N rates rose (Fig. 2).

SPAD readings are increasingly used as a quick and non-destructive method to monitor crop N content, growth status, and to predict grain yield. Maiti & Das (2006) reported that a SPAD reading of 37 as the threshold works well for N fertilizer application in wheat. There is a significant correlation between the chlorophyll measurement obtained using a SPAD meter and leaf N content in maize (Yu, Wu & Wang, 2010). Cabangon, Castillo & Tuong (2011) and Xiong et al. (2015) determined that there was a close relationship between leaf SPAD readings and leaf N content per leaf area. (Arregui et al., 2006) reported that the absolute or normalized relationships between the relative yield and grain N content with SPAD meter readings following a quadratic model and the Cate-Nelson statistical procedure. Monostori et al. (2016) indicated that SPAD readings could be used to predict wheat yield. Zhou & Yin (2017) determined that SPAD readings may be used to assess cotton N nutrition status and estimate cotton biomass. Edalat, Naderi & Egan (2019) reported that the combination of leaf N and SPAD data may become a tool to manage corn field N status and predict grain yield. A SPAD reading of 38 could be used to optimize N management by maintaining high grain yield and reducing N input by approximately 25% N (Cabangon, Castillo & Tuong, 2011), which was similar to our results.

Low N use efficiency may be caused by excessive N applications coupled with the inefficient splitting of N doses (Singh et al., 2010). The global N demand is expected to increase by 100–110% relative to 2005 usage levels, and the use of N is projected to increase from 100 Mt to 225–250 Mt by 2050 (Tilman et al., 2011), resulting in increased N losses, decreased economic benefits, and a decline in environmental health (Peng et al., 2006). It is important to determine the appropriate N rate for sustainable and productive rice production (Ku et al., 2016). N management practices should be easy to use, reliable, efficient, and environment friendly. SPAD readings would allow farmers to routinely monitor the leaf N status, adjust fertilization rates and times, avoid excessive N application, reduce N losses, and enhance N use efficiency.

The SPAD meter is a useful tool to determine plants’ N status but it might have limitations. The SPAD readings are taken from a small leaf area (six mm2) and require many repetitions before ascertaining the leaf N status (Rorie et al., 2011; Wang et al., 2014). In contrast, the leaf N contents could be determined using the micro-Kjeldahl digestion and distillation method (Yang et al., 2003), which is a destructive sampling method but is more accurate. The SPAD reading could be affected by many factors like leaf thickness, growth stages, genotype, chloroplast movement, irradiance, and field-to-filed variability of soil N supply (Samborski, Tremblay & Fallon, 2009; Naus et al., 2010; Singh et al., 2010; Kalaji et al., 2017). The infestation of plants with diseases or insects may also affect the results of the SPAD meter (Singh et al., 2010). This equipment requires qualified personnel to take leaf samples following a rigorous methodology (Reyes, Correa & Zñiga, 2017) and the measurements may deliver incorrect data if the plants are deficient in nutrients other than N (Kalaji et al., 2017). Previous studies have shown that nutrient deficiencies influenced the photosynthetic yield of PSII (Redillas et al., 2011; Msilini et al., 2013; Kalaji et al., 2014). It was suggested that the chlorophyll fluorescence could be used as a noninvasive method to detect nutrient deficiency that is more precise and sensitive (Goltsev et al., 2016; Kalaji et al., 2018; Horaczek et al., 2020).

Conclusion

Our study determined the positive effects of N rates on rice leaf SPAD readings, which can help determine more precise N application rates. The recommended N rates of OPT decreased by 5.8% and 15.3% in comparison with FP in the two experiments, respectively, based on leaf SPAD readings, when compared with FP doses. Similarly, the NAE of OPT increased by 5.8% and 10.0%, while PFPN increased by 6.0% and 14.2%, respectively, compared with FP. The SPAD meter allows farmers to monitor rice N status routinely, rapidly, and accurately. This may improve the timing and dose of N applications, reduce N losses, and improve N use efficiency.

Supplemental Information

Supplemental Information 1 Raw data

Click here for additional data file.

Additional Information and Declarations

Competing Interests

Author Contributions

Field Study Permissions

Data Availability

The authors declare there are no competing interests.

Wenfeng Hou and Weitao Xu analyzed the data, prepared figures and/or tables, and approved the final draft.

Juan Shen and Qiang Gao conceived and designed the experiments, performed the experiments, authored or reviewed drafts of the paper, and approved the final draft.

Muhammad Rizwan Khan, Yin Wang, Xue Zhou and Behzad Murtaza analyzed the data, authored or reviewed drafts of the paper, and approved the final draft.

Zhongqing Zhang performed the experiments, authored or reviewed drafts of the paper, and approved the final draft.

The following information was supplied relating to field study approvals (i.e., approving body and any reference numbers):

The fieldwork was conducted in privately-owned land with the permission of the farmer Jingfa Qu. We provided the seeds, pesticides and fertilizers, and the farmer was responsible for field management. We gave the grain to the farmer except for the essential samples.

The following information was supplied regarding data availability:

The raw data are available in the Supplementary File.

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
