# Peer review of "Recommended nitrogen rates and the verification of effects based on leaf SPAD readings of rice"

_PeerJ, doi:10.7717/peerj.12107_

## Round 0.1 · original submission · Major Revisions

The authors are requested to revise the MS fundamentally.

·

Basic reporting

need native English

the idea is clear, and the authors follow the constructions of the journal

the hypotheses is clear

the authors need to folow this article
A comparison between different chlorophyl lcontent meters under nutrient deficiency
conditions . Kalaji et al 2017 Journal of Plant Nutrition 2017, VOL. 40, NO. 7, 1024–1034

Experimental design

Two field trials comprising of seven nitrogen (N) rates (0, 45, 90, 135, 180, 225 and 270 kg N ha-1) were conducted in 2019 (two different fields in the same area with same planting pattern), using a randomized complete block design with three replicates

Validity of the findings

All underlying data have been provided; they are robust, statistically sound, & controlled
Conclusions are well stated, linked to original research question

Additional comments

the author need to follow (Kalaji et al 2017)
A comparison between different chlorophyl lcontent meters under nutrient deficiency
conditions . Journal of Plant Nutrition 2017, VOL. 40, NO. 7, 1024–1034

need improve your discussion part ,
answer the question (what is relationship between N and chlorophyll content

do you think it will give you same results and stress conditions?
can you compare your result with normal chlorophyll measurements?

Reviewer 2 ·

Basic reporting

Eglish level is poor and should be consalted with native speaker . Authors discribe the problem of N -fertilization only from one side. They see only the problem of over fertilizatision but they didn't saw that very often problem of insufficient fertilization which cause a problem of N deficiency

Experimental design

There is lack of information about kind of soil the experimental sites was localised there is no information about vegetation period characteristic and

Validity of the findings

All invetigation which authors done based on single basic method . There is suggest to compare gained results with some references methods. The authors did not provide information on the number of leaves in the sample for measurment

Additional comments

Rewrite the text using the suggestion in the review to improve qality of article

Annotated reviews are not available for download in order to protect the identity of reviewers who chose to remain anonymous.

---

## Round 0.2 · Major Revisions

Please read the cited paper of Kalaji et al. 2017: A comparison between different chlorophyll content meters under nutrient deficiency conditions.

You should add in the conclusion section that: such devices (like SPAD) allow non-invasive and trustable measurements of relative chlorophyll content. However, per Kalaji et al. 2017 such types of measurements may deliver false data if the plants are suffering due to deficiency of other nutrients than nitrogen.

English language should be revised by fluent English speakers.

·

Basic reporting

Using SPAD method, Easy to handle, reliable, high-efficiency, and environmentally friendly method to diagnose
plant nitrogen (N) status , predict grain yield and optimize N management is essential in
modern rice production systems

Experimental design

Original primary research within Scope of
the journal. Research question well defined, relevant
& meaningful. It is stated how the research fills an identified knowledge gap.
Methods described with sufficient detail &
information to replicate.

Validity of the findings

novality is poor

he must compare with normal analysis to chlorophyll or Nitrogen
to see what is difference between his method and normal method

Additional comments

i asked you to compare between normal analysis and SPAD even with previous studies

Reviewer 2 ·

Basic reporting

Language was improve and I don't see any mistakes, Literature was replenished with suggested possitions. Authors corect the manuscript with all sugested areas

Experimental design

Experiment design is better described than it was previously. Rsearch and statistical methods were now corectly described . Methods were descibed with sufficient details and information.

Validity of the findings

Conclusions are well stated, linked to original research question. Gained results are interesting for local perspective

Additional comments

Good job . In my opininion I can accept the manuscript and all changes which were done by authors

---

## Round 0.3 · accepted · Accept

Your article is now Accepted

Reviewer 3 ·

Basic reporting

The article "Recommended nitrogen rates and the verification of effects based on leaf SPAD readings of rice" is an interesting article, which can be accepted after considering following comments

1) line 54 "SPAD may reflect the plant leaves’ N status and is simple, quick, reliable, and harmless" Please explain SPAD more clearly.

2) Line 109 Please write the name of SPAD producer, company name, city, country.

3) Line 115 please indicate the years

4) Use Fig. or Figure - please check journal suggestion

5) Figure 2 legend is wrong "Temperature and precipitation during the rice-growing stage" please correct it

6) Xiong et al. (2015; https://doi.org/10.1038/srep13389) has shown that SPAD-based leaf nitrogen estimation is impacted by environmental factors and crop leaf characteristics, how the authors considered this issue please specify.

7) Manuscript can be further benefitted by following references
"https://doi.org/10.1626/pps.17.81"

Experimental design

No comments

Validity of the findings

SPAD meter is for a long time use for nitrogen estimation with some limitations please specify how your work is different, and not just a confirmatory work?